# Combined Dendritic and Axonal Deterioration Are Responsible for Motoneuronopathy in Patient-Derived Neuronal Cell Models of Chorea-Acanthocytosis

**DOI:** 10.3390/ijms21051797

**Published:** 2020-03-05

**Authors:** Hannes Glaß, Patrick Neumann, Arun Pal, Peter Reinhardt, Alexander Storch, Jared Sterneckert, Andreas Hermann

**Affiliations:** 1Section for Translational Neurodegeneration “Albrecht Kossel”, Department of Neurology, Universitätsmedizin Rostock, 18057 Rostock, Germany; hannes.glass@med.uni-rostock.de; 2Division of Neurodegenerative Diseases, Department of Neurology, Technische Universität Dresden, 01307 Dresden, Germany; neumannp90@gmail.com (P.N.); Arun.Pal@uniklinikum-dresden.de (A.P.); 3Department of Neurology, Städtisches Klinikum Dresden, 01129 Dresden, Germany; Jared.Sterneckert@tu-dresden.de; 4Center for Regenerative Therapies Dresden (CRTD), Technische Universität Dresden, 01307 Dresden, Germany; Peter.Reinhardt@abbvie.com; 5AbbVie Deutschland GmbH & Co KG, Neuroscience Discovery, Knollstrasse, 67061 Ludwigshafen am Rhein, Germany; 6Department of Neurology, University of Rostock, 18147 Rostock, Germany; Alexander.Storch@med.uni-rostock.de; 7German Center for Neurodegenerative Diseases (DZNE) Rostock, 18147 Rostock, Germany

**Keywords:** organelle trafficking, human induced pluripotent stem cells (iPSC), Chorea Acanthocytosis (ChAc), microfluidic chambers (MFCs), lysosomes, mitochondria, motoneurons (MN)

## Abstract

Chorea acanthocytosis (ChAc), an ultra-rare devastating neurodegenerative disease, is caused by mutations in the *VPS13A* gene, which encodes for the protein chorein. Affected patients suffer from chorea, orofacial dyskinesia, epilepsy, parkinsonism as well as peripheral neuropathy. Although medium spinal neurons of the striatum are mainly affected, other regions are impaired as well over the course of the disease. Animal studies as well as studies on human erythrocytes suggest Lyn-kinase inhibition as valuable novel opportunity to treat ChAc. In order to investigate the peripheral neuropathy aspect, we analyzed induced pluripotent stem cell derived midbrain/hindbrain cell cultures from ChAc patients in vitro. We observed dendritic microtubule fragmentation. Furthermore, by using in vitro live cell imaging, we found a reduction in the number of lysosomes and mitochondria, shortened mitochondria, an increase in retrograde transport and hyperpolarization as measured with the fluorescent probe JC-1. Deep phenotyping pointed towards a proximal axonal deterioration as the primary axonal disease phenotype. Interestingly, pharmacological interventions, which proved to be successful in different models of ChAc, were ineffective in treating the observed axonal phenotypes. Our data suggests that treatment of this multifaceted disease might be cell type and/or neuronal subtype specific, and thus necessitates precision medicine in this ultra-rare disease.

## 1. Introduction

Chorea Acanthocytosis (ChAc) is a rare neurodegenerative disease that affects 500–1000 persons worldwide (OMIM ID: 200150) [1]. It is part of the group of neuroacanthocytosis syndromes, which include McLeod syndrome, Huntington disease-like syndrome 2 and pantothenate kinase-associated neurodegeneration. The inheritance pattern is autosomal recessive and more than 134 causative mutations of ChAc are located in the locus of *VPS13A* (9q21.2) [2]. Patients affected by the disease present various clinical features including orofacial dyskinesia and choreatiform movements with typical drops of the upper body. Most patients of the disease share signs of peripheral neuropathy. Other features are irregular but may include epilepsy in the early stages and Parkinsonism as the disease progresses. The affected neurons mainly responsible for the choreatiform movement disorder are medium spiny neurons (MSN) of the basal ganglia [3]. However, many other neuronal subtypes are affected as well, such as dopaminergic neurons or motoneurons, explaining the plethora of maladies including parkinsonism and/or peripheral neuropathy [4].

The affected gene locus is conserved in higher organisms and various functions have been attributed to its product in lower organisms. Pioneering work on these functional aspects was conducted in yeast, where vacuole sorting, sporulation and inter organelle signaling was described [5,6]. More recently, further functional aspects of VPS13A were revealed by description of its function as a lipid transport protein, where it is located at various membrane contact sites—most notably the endoplasmic reticulum and mitochondria [7,8,9,10].

Structural analysis showed that Vps13 has an APT1 and SHR_BD domain that is responsible for interaction with lipids, among which PI3P is pivotal, due to its ability to guide organelles and facilitate the recruitment of other proteins [11].

The main focus of interest in the scientific community so far has been the prominent feature presented by the acanthocytes and thus the molecular understanding of the disease mainly arose from work in red blood cells or related hematic models. Current understanding is highlighting two major pathways being affected. Changes in the activity of PI3K and the subsequent kinases RAC1 and PAK1 were shown to have a strong effect on the polymerization of cortical actin, hence causing an acanthocytic shape in erythrocytes [12,13,14]. Also downstream of PI3K is the serum and glucocorticoid inducible kinase SGK1, which regulates the activity of NFκB signaling, ultimately regulating the Ca^2+^ channel subunit Orai1 and store operated Ca^2+^ entry (SOCE). Hence, it was shown that chorein deficiency resulted in lower Orai1 expression and reduced SOCE, which was at least partially restored upon lithium treatment [15,16]. This finding was later confirmed in medium spiny neurons [17].

The second pillar of the underlying pathophysiology is the SRC-kinase Lyn. Upon depletion of chorein, Lyn becomes hyper active and subsequently phosphorylates its targets, including the anion transport protein SLC4A1 (band 3) [18]. This was shown to be the major driving force in impairing autophagy [19]. Furthermore, the hyperactive Lyn was shown to be responsible for the hyperexcitability, which was reported in ChAc patient induced pluripotent stem cell (iPSC) derived MSN [20]. Treatment of affected cell cultures with the SRC-kinase inhibitor 4-Amino-5-(4-chlorophenyl)-7-(dimethylethyl)pyrazolo(3,4-d)pyrimidine (PP2) reverted the phenotypes back to wild type levels.

In our recent study on human patient-derived MSNs, however, we were able to show specific neuronal impairments of mitochondrial and lysosomal morphologies and trafficking features, which were not affected by Lyn kinase inhibition using PP2 pointing towards cell type specific effects [21]. The current study focuses on the effects of chorein deficiency in patient iPSC derived spinal motoneurons, the cell type causing the hallmark symptom of peripheral motor neuropathy.

## 2. Results

### 2.1. Loss of VPS13A Function Does Not Interfere With Midbrain/Hindbrain Differentiation Capacity

The first objective of this work was to investigate the differentiation capacity of the *VPS13A* mutants and respective wild type controls (Ctrl). To this end, we subjected previously obtained patient specific small-molecule neuronal progenitor cells (smNPCs) to a differentiation protocol yielding midbrain and hindbrain neurons. We investigated different protein markers by using immunofluorescence (IF)-staining to distinguish between different stages of differentiation. SOX2 is a transcription factor typically expressed in NPCs; TUBB3, a neuron specific member of the tubulin-family, highlights immature neurons; MAP2 was used to mark mature neurons; tyrosine hydroxylase (TH) is the rate limiting enzyme for biosynthesis of dopamine and is used to detect dopaminergic neurons; SMI-32 is a neurofilament subclass typically found in motoneurons; GFAP and GALC are used to identify astrodendroglia and oligodendroglia cells, respectively (Figure 1a). At the stage of neural stem cells and immature neurons, no differences were found between Ctrl and ChAc lines for investigation SOX2 (Ctrl: 53.0%, ChAc: 46.79%, *p* = 0.55) or TUBB3 (Ctrl: 12.8%, ChAc: 17.7%, *p* = 0.37). Generation of mature neurons was not impaired either, as identified by staining of MAP2 (Ctrl: 29.9%, ChAc: 30.9%, *p* = 0.92). The amount of mature dopaminergic neurons did not differ between Ctrl and ChAc, as shown by TH/MAP2 double-positive cells (Ctrl: 52.3%, ChAc: 69.3%, *p* = 0.23; Figure 1b). Since the employed protocol was only capable of yielding low amounts (< 1%) of GFAP positive astrocytes or GALC positive oligodendrocytes, we did not systematically analyze these. The anti-SMI32 antibody (detecting spinal motoneurons) that we applied did not detect peri nuclear protein. We therefore decided to quantify the whole network size. No difference was observed in the size of the MAP2^+^ network or the SMI32^+^ network (MAP2^+^: *p* = 0.48; SMI32^+^: *p* = 0.63; Figure 1c). Their overall extension was similar. Thus, we showed robust differentiation to mature midbrain and hindbrain neurons, which was not affected by mutations in the *VPS13A* gene, fitting to the nature of a neurodegenerative disease.

Eager to probe for first hints of neurodegeneration, we employed an IF-based assay to quantify the fragmentation of the MAP2^+^ and SMI32^+^ neurites. The length of the network was assessed by skeletonizing the neuronal network to one pixel wide lines and measuring the number of pixels. The calibration factor of the image was then used to calculate the extension. The fragmentation of the skeletonized network is an indicator for disrupted neurites (Figure 1d; For further explanation of this method, please refer to Figure A1). Interestingly, SMI32^+^ neurites showed an overall increased fragmentation compared to MAP2, but there was no difference observed between Ctrl and ChAc patients’ cell lines (*p* = 0.46). Analysis of MAP2 network, however, revealed a small but significant increase in fragmentation, suggestive of an increased degeneration of ChAc patients’ neurons (*p* = 0.0022). In order to gain a better understanding of the extent of degeneration, cell survival after prolonged culture (6 months) was measured and we found a significant decrease of surviving ChAc neurons (*p* = 0.0089; Figure 1e). This is in accordance with our recently described neurodegenerative phenotype of ChAc patient derived MSNs and shows that the observed degeneration is not only present in the primarily affected neuronal sub population, but might be a general phenomenon [20].

### 2.2. Midbrain/ Hindbrain Neurons of ChAc Patient’s Show Altered Growth Characteristics

Since peripheral neuropathy is a main symptom of ChAc patients, which is particularly helpful to distinguish it from diseases such as Huntington’s disease, we next looked for neurite outgrowth of individual neurons within the first 72 h after seeding following the neuronal induction. As expected, the total length of all neurites significantly increased over time. However, there was no difference observed when we compared Ctrl and ChAc neurons (Figure 2a). The average number of branching points per neuron was not affected by time or genotype (Figure 2b). Interestingly, when we focused solely on the longest neurite—most commonly the axon—, we saw significant longer processes in ChAc neurons (Figure 2c). Since the total length and branching points were unaffected, we further quantified the number of primary neurites projecting from the cell soma. Indeed, ChAc patients’ neurons had on average significantly fewer primary neurites (Figure 2d).

### 2.3. Micro Fluidic Chambers Allow for a Selective Directional Growth of Motoneurons

We sought to establish a more functional readout for this outgrowth phenotype and used micro fluidic chambers (MFCs) to manipulate the growth factor gradients and therefore enforce a directional growth rather than observing random trajectories [22]. First, we analyzed which neuronal subtype is mainly sprouting through the MFCs. There were almost no TH-positive neurites observed, and, almost all neurites were positive for the motoneuron marker SMI-32 (Figure 2e). Strikingly, we discovered that MFCs also allowed us to investigate axon selectively, when we observed that the distal site was void of MAP2^+^ dendrites (Figure 2f). This enabled us to use this assay to study selectively axonal motoneuron growth. We analyzed the rate of the growth-through channels by manually counting the fully penetrated channels (exit sites) over the course of 23 days. While there was no difference when brain-derived neurothrophic factor (BDNF) was used as the growth cue, ChAc motoneurons showed a clearly diminished growth rate in the absence of BDNF (Figure 2g,h).

### 2.4. Midbrain/Hindbrain-Derived Neurons Highly Express NTRK2

In the family of neurotrophic tyrosine kinase receptors, the tropomyosin receptor kinases A, B and C (NTRK1-3) are able to bind BDNF and the structurally related neurotrophic factors NGF, NT-3 and NT-4. While NTRK1 has a high affinity for the nerve growth factor (NGF) and NTRK3 has a high affinity for NT-3, NTRK2 recognizes BDNF, NT-4 and NT3 [23]. Furthermore, it was shown that in Huntington’s disease patients’ neurons, the retrograde transport of NTRK2, and therefore BDNF signaling, is impaired by mutated HTT [24]. Since Huntington and ChAc share common features, we were interested in whether NTRK2 is involved in the disease pathogenesis of ChAc as well. When expression patterns of *NTRK1*, *NTRK2* and *NTRK3* were analyzed, *NTRK2* was significantly more highly expressed than the other two receptors. (*NTRK2* vs. *NTRK1*: *p* < 0.0001; *NTRK2* vs. *NTRK3: p* < 0.0001; Figure 2i). After finding a BDNF dependent phenotype and a robust expression of its receptor *NTRK2*, investigation of whether the expression levels of *NTRK*s were different in ChAc patients’ midbrain/hindbrain-derived neurons compared to Ctrl ones was conducted. Even though no significant differences were found, the observed trend of reduced expression of *NTRK2* in ChAc patient-derived neurons was close to being significant. (*NTRK1*: *p* = 0.13; *NTRK2*: *p* = 0.06; *NTRK3*: *p* = 0.13; Figure 2j).

### 2.5. Tubulin Network is Unaffected in ChAc Midbrain/Hindbrain Cell Cultures

It is known that chorein deficiency leads to altered cytoskeletal organization in the form of an increased G/F-actin ratio in erythrocytes and striatal neurons, but not in midbrain/hindbrain neurons [20]. To further shed light on the ongoing changes, we investigated the ratio of a different, equally important protein—namely tubulin. However, upon isolation and western blot analysis, no differences in the ratio of G/F-tubulin were observed (*p* > 0.99; Figure 2k). Since tubulin modification plays a pivotal role in its utility and stability, we probed for one of the most common modifications of tubulin, its acetylation. Upon confirming the absence of any difference in the stability related G/F-ratio assay, no differences in the tubulin-acetylation-rate was observed when Ctrl and ChAc midbrain/hindbrain neuronal cell cultures were compared (*p* > 0.66; Figure 2k).

### 2.6. ChAc Patients’ Motoneurons Harbor Altered Mitochondria

Since the growth of neuronal processes requires extensive cytoskeletal remodeling and therefore creates a high-energy demand that has to be met by the cell, we sought to investigate whether the observed impaired outgrowth capacity of ChAc motoneurons are symptoms of reduced capability of their mitochondria. First, we assessed the number of mitochondria, their morphology, as well as their potential in proximal and distal parts of the axons. To this end we utilized live cell imaging with living dyes for mitochondria (MitoTacker DeepRed; MitoProbe JC-1) to analyze morphologic parameters of the organelles from the first image of each video stack as well as properties of their movement (Figure 3a). The number of mitochondria in ChAc patient-derived neurons was significantly reduced, when compared to their Ctrl counterpart (*p* = 0.0003; Figure 3b). The shape of these remaining mitochondria can be described using the aspect ratio of ellipses, which were fitted on them. The aspect ratio is then defined as the ratio between the larger and smaller diameter of this ellipse. ChAc patients’ neurons’ mitochondria were shortened, as was measured by a significant reduction of their aspect ratio (*p* < 0.0001; Figure 3c). The overall size of these mitochondria was reduced as well, as measured by the pixel area occupied by organelles (*p* = 0.0042; Figure 3d).

The application and quantification of tracking algorithms showed that the fraction of moving mitochondria of neurons from ChAc lines was moderately increased (*p* = 0.0017; Figure 3e), while their mean velocity was reduced (*p* < 0.0001; Figure 3f). We then went on to investigate the processivity of mitochondria movements. This parameter was calculated by dividing the displacement of each track—the distance between its start and end point—by the total length of the track. A highly processive moving track goes straight from its start to its end point, while an improcessive track may stop or even revert its direction, therefore adding to its total track length while not changing the track displacement. Interestingly a ChAc patient’s mitochondria showed a reduction of processivity (*p* < 0.0001; Figure 3g).

### 2.7. Motoneurons and Dopaminergic Neurons Have Distinct Mitochondrial Phenotypes

To further investigate whether above-mentioned phenotypes reflect abnormalities of the motoneuron cell population of the mixed culture and to analyze axonal changes specifically, we utilized our MFC culture system to investigate selectively motoneurons (Figure 2e). Besides the separation of these neuronal subpopulations, the system further enabled us to investigate proximal and distal parts of the axons independently and gave us the opportunity to identify the direction of moving organelles (Figure 4a,b). While anterograde movement describes transport to the synaptic terminals, retrograde movement is headed to the cell’s soma. Morphological analysis—as done before in unguided cultures—revealed that the number of mitochondria in MFCs was not significantly different between ChAc and Ctrl in proximal or distal positions (Figure 4c). This finding contrasts with the result obtained in unguided cultures and either hints that the observed phenotype in the unguided cultures is represented mainly by dopaminergic neurons or that—because of the nature of these cultures—is dominated by dendritic phenotypes. Analysis of the aspect ratio revealed a significant difference between ChAc neurons’ mitochondria and Ctrl ones in the distal position (Figure 4d) fitting to the above-discovered shortening of mitochondria in unguided cell cultures.

Statistical analysis on the data of moving tracks did not identify significant differences (Figure 4e). When the mean speed of the moving fraction was evaluated by 2-way ANOVA, we found a significant reduction of the mitochondria speed in the distal axon compared to the proximal one (Figure 4g). Post-hoc tests further revealed a significant increase in the mean speed at the proximal site in ChAc derived motoneurons when compared to Ctrl ones. This is in striking contrast with the unguided cultures, in which we observed a reduction of speed in ChAc neurons.

MFC cultures gave us the opportunity to investigate the direction of mitochondrial movement—either anterograde (direction to synapses) or retrograde (direction to nucleus). 2-way ANOVA revealed significant influences of the genotypes, readout position as well as interaction between both. Post-hoc analysis then highlighted a reduction in the amount of anterograde trafficking mitochondria in proximal parts of axons of ChAc patient derived neuronal cultures in comparison to Ctrl ones (Figure 4f).

When analyzing the mean speed of the anterograde transport with 2-way ANOVA, the only difference observed was in the comparison of distal axonal with proximal axonal readout sites. There was, however, no difference between ChAc and Ctrl cell lines. The mean speed of retrogradely moving mitochondria was significantly lower in the distal parts than in the proximal ones as well. Furthermore, in the proximal readout field-of-view mitochondria in ChAc patient derived neurons moved significantly faster (Figure 4h). The aforementioned increase in speed of ChAc patient neurons’ mitochondria, therefore, can be explained by an increase in the retrograde speed. Both findings of more and a faster retrograde transport of mitochondria hint at mitochondrial impairment, which in turn necessitates an increase in retrograde transport for perinuclear mitochondrial recycling.

In addition, 2-way ANOVA of the processivity revealed that mitochondria move in a less processive way in distal axonal sites than in proximal ones. Post-hoc analysis showed that in proximal sites, the processivity of moving ChAc patient-derived neurons’ mitochondria was significantly increased. (Figure 4i). Separate analysis of anterograde and retrograde transport yielded a difference in both anterogradely and retrogradely moving fractions alike (Figure 4j).

### 2.8. Mitochondria of ChAc Derived Mature MN Show a Strong Distal Hyperpolarization

In order to look for other signs of mitochondrial damage, we investigated their membrane potential with the live cell dye JC-1 (Figure 4k). If a mitochondrion has a high membrane potential, JC-1 forms a homodimer and emits red fluorescence, while a low membrane potential emits green fluorescence. Therefore, polarization of each mitochondrion was evaluated by calculating the ratio of the red and green fluorescence, when stimulated with the same wavelength and intensity. Strikingly, the signal in the distal axonal sites of mature ChAc neuronal cultures shifted tremendously towards red fluorescence, indicating a strong hyperpolarization (Figure 4l).

### 2.9. Loss of VPS13A Function Induced Altered Lysosomal Phenotypes

Another group of organelles, which is pivotal for homeostasis of neurons, is comprised of endo-lysosomal vesicles. To visualize these compartments, cells were incubated with LysoTracker Red DND-99 prior to live cell imaging, which enabled us to investigate endo-lysosomal vesicles under the same conditions as the previous MitoTracker-based experiments (Figure 5a).

The morphological examinations in mixed cultures yielded a significant increase in the normalized number of lysosomes in ChAc patients’ neurons (*p* = 0.018; Figure 5b). Their shape and size remained unaltered compared to Ctrl cell lines (Figure 5c,d). We characterized two parameters of the motility of endo-lysosomal compartments in unguided cell cultures. The number of the moving fraction was unaltered in mature neurons from ChAc patients (Figure 5e). Interestingly, the mean velocity was significantly reduced in ChAc neurons (*p* < 0.006; Figure 5f). Processivity of moving lysosomes is in general higher than of moving mitochondria and showed a significant reduction of directed movement of lysosomes in ChAc patients’ neurons compared to Ctrl ones (*p* = 0.0061; Figure 5g).

Following the same approach, as was used with MitoTracker, MFCs were used to differentiate between proximal and distal parts of axons, while selectively monitoring motoneurons. Morphological analysis of lysosomes in axons of neurons grown in MFCs was carried out. Statistical analysis showed no difference in the observed normalized number of lysosomes between ChAc patients’ neuronal cultures and Ctrl ones (Figure 6a). When we studied the aspect ratio of LysoTracker^+^ compartments, a significant difference between distal axonal sites and proximal ones was observed. Post-hoc tests revealed an increase in the aspect ratio in proximal sites of ChAc patient derived neurons’ lysosomes in comparison to Ctrl ones, suggesting different cargo loading (Figure 6b).

The fraction of moving endo-lysosomal vesicles of neurons derived from midbrain/hindbrain differentiation was higher in distal axonal sites compared to proximal ones, as revealed by 2-way ANOVA (Figure 6c). There was, however, no difference between ChAc lines and Ctrl lines. Statistical analysis of mean speed data obtained from MFCs reported no significant differences between ChAc patient derived neurons’ lysosome mean speeds and Ctrl ones (Figure 6d). This is in contrast to the results obtained in unguided cell cultures and—similar to the observation of the mean velocity of mitochondria—might be attributed to the selective investigation of motoneurons or inherent differences of the cell culture systems. When the tracks were analyzed in detail regarding their movement direction, statistical analysis highlighted no significant differences (Figure 6e). Statistical analysis of the mean speed data of anterograde movement of lysosomes reported no significant differences (Figure 6g). When the same analysis was conducted on the data of the mean speed of the retrograde moving fraction, however, we identified a significantly lower speed in the distal axonal sites compared to the proximal ones. Post-hoc analysis also revealed an increased speed of retrogradely moving lysosomes in the proximal axonal sites of ChAc patient derived neurons’ compared to Ctrl ones. This increase could suggest a higher demand for endo-lysosomal activity.

We quantified processive movement of lysosomes in neurons grown in MFCs, however, statistical analysis revealed no significant differences (Figure 6e). There was no difference observed as well, when we separately analyzed the data of anterogradely and retrogradely moving lysosomes (Figure 6h).

### 2.10. ChAc Midbrain/Hindbrain Neuronal Cultures’ T Trafficking Profile is Different to the One from MSN Cultures

After observing various differences between ChAc-patient and Ctrl-derived neuronal cultures, we sought to obtain a comprehensible and intelligible representation for the observed morphological and trafficking readouts. To this end, a high content disease profile was generated and compared to a previously obtained MSN-derived profile of ChAc patient-derived lines [21]. The depicted signatures show clearly distinct profiles (Figure 7). The major phenotypes manifest in mitochondria, while the differences observed in the endo-lysosomal fraction are noticeable but smaller. Furthermore, these mitochondrial trafficking differences occur in the proximal sites. The phenotypic signatures of midbrain/hindbrain- and MSN-derived neurons are different in their intensity but show a similar pattern, which suggests that the underlying pathomechanistic processes are very likely similar in both neuronal sub types.

### 2.11. Inhibition of Lyn-Kinase Showed No Effect on ChAc Neurons’ Phenotypes

After thoroughly characterizing the organelle phenotypes, we sought to investigate how the mature midbrain/hindbrain neuronal cultures of ChAc patients respond to Lyn kinase inhibition, which was shown to ameliorate phenotypic distortions in ChAc models [20]. To this end, mature neuronal cultures of ChAc patients and controls were incubated with 10 µM PP2 for 48 h, a well-known Srk-kinase inhibitor, and morphology and trafficking of mitochondria and lysosomes of neurons in unguided cultures was investigated. Similar to results obtained in MSN, when we investigated mitochondria of treated ChAc patients’ neuronal cultures, we did not see any effect in the normalized number, aspect ratio or size (Figure 8a–c). Investigation of the moving fraction of mitochondria showed an increase of PP2 treated ChAc patients’ neurons’ mitochondria compared to untreated conditions. this effect was solely observed in ChAc cell cultures but not in Ctrl ones, which ultimately led to significantly more motile mitochondria in PP2 treated ChAc derived neurons compared to PP2 treated Ctrl ones (Figure 8d). Statistical analysis on the mean speed of moving mitochondria reported no significant difference (Figure 8e). When the processivity of moving mitochondria was analyzed, a slight, yet significant reduction in PP2 treated ChAc patients’ neurons’ mitochondria compared to untreated ones was present upon incubation with PP2 (Figure 8f). Further analysis of the post-hoc tests showed that the processivity was significantly reduced compared to Ctrl lines as well (Figure 8f). Analysis of lysosomal organelles gave a similar overall result. While there was no effect in the normalized number and size of lysosomes, the aspect ratio was altered (Figure 8g–i). Under sham conditions, ChAc patients’ neurons’ lysosomes were significantly rounder than Ctrl ones, while upon PP2-treatment they were not. This was mainly driven, however, by changes of the Ctrl lysosomes. Post-hoc analysis revealed that the reduction in the aspect ratio of PP2 treated Ctrl neurons’ lysosomes was significant compared to untreated ones (Figure 8h). Statistical analysis of data regarding the moving lysosome fraction showed no significant difference (Figure 8j). When calculating a 2-way ANOVA of the data on the mean speed of moving lysosomes, PP2 treatment significantly reduced the mean speed in both Ctrl and ChAc cell cultures. However, post-hoc analysis showed no differential effect on Ctrl and ChAc derived neurons’ lysosomes (Figure 8k). The calculated 2-way ANOVA of the processivity data on lysosomal organelle movement reported a significant influence of the PP2 treatment as well. Upon investigation of the post-hoc tests, we found that the processivity of PP2 treated Ctrl neurons’ moving lysosomes was significantly lower when compared to DMSO-sham (Figure 8l). We were not able to identify a significant difference in comparison to ChAc patients’ neurons moving lysosomes, though. In summary, the 48 h treatment with PP2 had little to no effect on the morphology and trafficking characteristics of mitochondria and lysosomes of ChAc patients’ cell lines as well as Ctrl ones.

## 3. Discussion

Major advances in understanding the pathomechanism of ChAc and the function of chorein and its orthologues have been made recently [7,8,9,10,25]. Still, the main focus of the scientific community lies on the investigation of erythrocyte and medium spiny neuron (MSN) phenotypes as the primarily affected cell types. ChAc patients suffer from a variety of maladies, of which peripheral (motor) neuropathy is a main symptom. In this article, we comparatively investigate for the first time the influence of chorein deficiency on dopaminergic neurons and spinal motoneurons, to gain a better understanding on the late stage ChAc, which can present with Parkinsonism as well as peripheral neuropathy.

We were not able to identify any abnormalities in the differentiation capacity of ChAc iPSC into midbrain/hindbrain neurons, which is in agreement with previous publications that showed a normal differentiation towards MSN [20,21]. In line with this observation, the age of onset and the clinical presentation of ChAc patients strongly suggests a degenerative progression rather than a developmental defect.

Using a MFC system to separate axons from dendrites as well as neurons from non-neuronal cells was documented before and is a well-established technique [22,26]. Motoneurons are particularly well suited to penetrate the long distance of the micro channels, because of their sensitivity to the chemotactic guidance of GDNF and BDNF and their nature of generating very long axons and growth cues. We were able to separate them from the other neuronal subtypes and non-neuronal cells using this culture system.

Interestingly, the growth of ChAc axons through the MFCs depends on BDNF substitution. This could either hint to a lack of the endogenous production of BDNF by the cells or a defective signaling which requires elevated BDNF levels to begin with. The concentration of BDNF used in this assay is frequently used in these culture systems [27]. BDNF levels in the brain are dependent on external stimuli. During context-dependent fear-conditioning tests in mice, the baseline BDNF concentration in the cerebrospinal fluid was estimated at 1–10 pg/mL [28]. Endogenous levels of BDNF in the human brain are not well documented, however, we believe that the herein used concentration is exceeding the average concentration and therefore should result in a constant activation of BDNF receptors. Since under these BDNF-excessive conditions, there was no ChAc-specific phenotype present and the expression levels of *NTRK1*, *NTRK2* and *NTRK3* were not different, we think the resulting effects are due to changes downstream in the BDNF signaling cascade.

We found that the mitochondria in unguided cultures share similar morphological features in midbrain/hindbrain neurons with mitochondria in MSN [21]. The trafficking phenotype of midbrain/hindbrain neurons was similar to the one seen in MSN as well. The striking hyperpolarization we observed in the distal axonal sites was reported in MSN as well [21]. We believe that an increased mitochondrion potential can also lead to neurodegeneration, as was reported in the literature [29,30]. Interestingly, it is also described as a measure to counter apoptosis [31]. Since all mitochondria of the distal axon have to pass proximal site, it remains unclear whether the observed shift in potential is upstream or downstream of the proximally observed trafficking phenotypes. Together with the previously described changes in mitochondria trafficking, especially the increase in retrograde transport, is very likely reflecting increased mitochondrial damage, which urges the cell to recycle it´s mitochondria—a feature, which is usually done in the perinuclear region. In summary, these results indicate that motoneurons suffer from mitochondrial damage in a similar pattern as observed in MSN. We believe that the differences are reflecting the divergent natures of these different neuronal subtypes.

The morphological analysis of endo-/lysosomes revealed an increase in number of organelles. In order to shed light if this is due to an overproduction of lysosomal vesicles or due a deficit in downstream fusion of these compartments, further experiments need to be conducted. Even though we did not comprehensively investigate the underlying mechanism, both possibilities suggest an impairment of this pathway, as was described previously [25,32]. The altered transport features of these compartments can be a secondary effect due to mitochondrial failure to provide enough ATP to the cells or it can be indicative that the underlying rails—the microtubules—are defective. However, we did not find a significant difference in microtubule polymerization as well as acetylation (Figure 2k). Nonetheless, it is still possible that other substantial changes in microtubule modification are present, e.g., phosphorylation, tyrosinylation or alterations of microtubule binding proteins such as Tau.

We identified differences for mitochondria as well as lysosomes during morphological and trafficking characterization between unguided cultures and MFCs. These could have different reasons. As mentioned above, the exclusive analysis of motoneurons in MFCs could be one reason for this, as well as unspecific factors due to the culture conditions per se. However, these differences could also point towards differences between axons and dendrites. The latter would fit to the data obtained in unguided cultures (Figure 1) showing a disease phenotype only in dendrites (MAP2 fragmentation) but not in axons (no differences in SMI32 fragmentation between ChAc and Ctrl.) and reduced amount of dendrites (Figure 2d) while having increased axon lengths (Figure 2c). However, further studies need to elaborate on this in more details.

The high content disease profile we generated for ChAc patients’ midbrain/hindbrain neurons indicated that different neuronal subtypes share common phenotypes, which are very distinct from other neurodegenerative diseases such as ALS [33]. The intensity of the phenotypes corresponds to the clinical severity of the mutations. While the previously published FUS-P525L mutation has a very early disease onset and a fast progression, ChAc has milder course and is usually later onset. In contrast to the ALS profile, which harbors a strong distal phenotype, which is sometimes referred to as “dying-back”, the more pronounced phenotypes in ChAc patient’s neurons were observed in the proximal axon. Even though the singular most striking phenotype of the distal axon was the hyperpolarization of mitochondria, we cannot be certain that this is not a downstream symptom of the mitochondrial impairments in proximal axons (Figure 7). We therefore believe that—given the current data set—the neurodegenerative progression of neurons affected by ChAc starts at their proximal site.

We were not able to revert the disease phenotypes back to control conditions by means of treatment with PP2. Some readouts suggested that the effect attributed to the carrier, DMSO, was larger than that of the Srk-kinase inhibitor (Figure 8c,d,f). We observed a similar behavior when investigating MSN previously [21]. Since the observed phenotypes resemble the ones we described in MSN, this further strengthens our believe that the underlying mechanism that leads to these subtle organelle impairments is mediated by a pathway different to Lyn-kinase or is indeed affected by parts which are further upstream. One other candidate for this different pathway is the PI3K signaling pathway, which was already investigated in the context of survival and cytoskeletal organization of erythrocytes [3]. Our findings suggest that the same pathomechanism affects different neuronal subtypes that are similar to MSN but manifest in different clinical symptoms—e.g., peripheral neuropathy for motoneurons and parkinsonism for dopaminergic neurons but might be different to other tissues. These findings highlight the complex requirements for an effective treatment of this multifaceted disease, which calls for taking clinical pathological phenotypes that are not concerned with the primarily affected neuronal sub populations into account.

## 4. Materials and Methods

### 4.1. Materials Patient Characteristics

Patient characteristics are shown in Table 1. All procedures were in accordance with the Helsinki convention and approved by the Ethical Committee of the Technische Universität Dresden (EK45022009).

### 4.2. iPSC Derivation

We generated iPSC from patients shown in Table 1. Derivation and characterization of these lines was described previously [20]. In short, we transfected fibroblasts using four monocistronic retroviral vectors carrying the Yamanaka factors—OCT4, SOX2, KLF4 and MYCC (addgene #17217, #17218, #17219, #17220, #8449, #8454). We cultivated emerging colonies on MMC (Tocris Bioscience, Bristol, UK)-treated mouse-embryonic-fibroblasts in embryonic stem cell (ESC)-medium (77.8% KO-DMEM (Thermo Fisher Scientific, Waltham, Massachusetts, USA), 20% KO-Serum (Thermo Fisher Scientific, Waltham, Massachusetts, USA), 1% non-essential amino acids (Thermo Fisher Scientific, Waltham, Massachusetts, USA), 1% Penicillin/ Streptomycin (Thermo Fisher Scientific, Waltham, Massachusetts, USA), 0.2% β-mercaptoethanol (Thermo Fisher Scientific, Waltham, Massachusetts, USA)), supplemented with bFGF (Sigma Aldrich, St. Louis, Missouri, USA). We then characterized of these lines by alkaline phosphatase staining, confirmation of exogene silencing/activation of endogenous transcription factors by a quantitative polymerase chain reaction, immunocytochemistry of embryonic stem cell markers, 3-germ-layer-differentiation and confirmation of ChAc-phenotype by chorein western blot. We used two clones from each individual in all of the following experiments. We subsequently differentiated the obtained iPSC lines into small-molecule neuronal-precursor-cells (smNPC).

### 4.3. Derivation of smNPCs

We conducted our experiments in a smNPC-derived neuronal cell model system, since they are shown to generate a reproducible amount of midbrain dopaminergic neurons as well as hindbrain specific motoneurons—which is ideal for our investigative scope of not-primarily affected neuronal subtypes in ChAc. We derived smNPC from iPSCs, as reported previously [22,26,28]. In short, we cleaned spontaneous differentiated colonies off adherent iPSC cultures and incubated with 2 mg/mL type IV collagenase for one hour. We then collected detached colonies and cultivated them as embryoid bodies in ESC-media supplemented with 1 µM Dorsomorphin (Tocris Bioscience, Bristol, UK), 10 µM SB-431542 (Tocris Bioscience, Bristol, UK), 3 µM CHIR99021 (Cayman chemical company, Ann Arbor, Michigan, USA) and 0.5 μM purmorphamine (PMA) (Cayman chemical company, Ann Arbor, Michigan, USA). We replaced the ESC-media with N2B27 (49% DMEM/F12, 48.5% NeuroBasal, 0.5% N2-supplement, 1% B27-supplement, 1% Penicillin/Streptomycin/L-glutamine) after 2 days. Dorsomorphin and SB-431542 were omitted from day 4 and 150 µM ascorbic acid (AA) (Sigma Aldrich, St. Louis, Missouri, USA) was additionally added. On day 6 we triturated the embryoid bodies and seeded them on matrigel (BD Bioscience, Franklin Lakes, New Jersey, USA) coated plates, which were prepared one day prior to thawing or passaging. We cultivated the lines until passage 10 times to obtain a stable smNPC phenotype, and this was confirmed by quantitative reverse transcription polymerase chain reaction qPCR.

### 4.4. Culture of smNPCs

We expanded smNPCs continuously on matrigel coated plates in N2B27 media supplemented with 3 µM CHIR99021, 150 µM AA and 0.5 µM PMA, which we changed every other day during proliferation phase. At each split we seeded 100,000 cell/cm². When confluence was reached after roughly one week, we passaged the cultures or initiated differentiation. We washed the cells twice with warm PBS and treated them for 10 min at 37 °C with accutase. The enzymatic reaction was quenched by adding double the volume of warm DMEM/F12 (Thermo Fisher Scientific, Waltham, Massachusetts, USA). We collected the cell suspension and centrifuged at 250 g for 5 min. We then discarded the supernatant and resuspended the cells in 1 mL of warm N2B27 media supplemented with 3 µM CHIR99021, 150 µM AA and 0.5 µM PMA. We assessed concentration by diluting the suspension 1:100 and counting the cells with a Neubauer slide.

### 4.5. Midbrain/Hindbrain Differentiation

We differentiated smNPC cultures into mature neurons using a multistep protocol according to Reinhardt et al. [26]. Briefly, we seeded smNPCs in a 6-well and changed media every other day. For the first 2 days, we cultivated cells under normal proliferation conditions using N2B27 supplemented with 3 µM CHIR99021, 150 µM AA and 0.5 µM PMA. On day 2 we changed medium to induction conditions and applied N2B27 supplemented with 200 µM AA, 1 µM PMA and 10 ng/mL FGF8 (R&D Systems, Minneapolis, Minnesota, USA). On day 10, we passaged cells and reseeded them into their final format. For ICC, we seeded cells in 4-well plates with coated cover slips at a density of 100,000 cells/well. For selective motoneuron analysis, we used MFC and injected 300,000 cells in one macro channel of each mounted silicon device. From then on, we fed the cultures every other day with maturation media consisting of N2B27 supplemented with 20 ng/mL BDNF (Promega, Madison, Wisconsin, USA), 10 ng/mL GNDF (Sigma Aldrich, St. Louis, Missouri, USA), 1 ng/mL TGFβ3, (Peprotech, Rocky Hill, USA) 0.5 mM dbcAMP (Sigma Aldrich, St. Louis, Missouri, USA) and 200 µM AA. To achieve growth-through in our MFCs, we applied a growth factor gradient. The “soma”-site was supplemented with 1 ng/mL of TGFβ3, 0.5 mM dbcAMP and 200 µM AA, while the “synapse”-site received a complete media including 20 ng/mL BDNF and 10 ng/mL GNDF. We cultivated the maturating neuronal cultures for three weeks before immunocytochemistry or live cell imaging.

### 4.6. Immunocytochemistry

For ICC analysis, we washed mature neurons once with PBS and then fixated them by incubation with pre warmed 4% PFA solution for 12 min. After fixation, we washed the cells three times with PBS and subjected them to permeabilization for 10 min at room temperature using following solution (0,2% Triton X-100 (Thermo Fisher Scientifc, Waltham, Massachusetts, USA) in PBS). After permeabilization, we applied blocking buffer for 1 h at room temperature. The blocking buffer contained: 1% bovine serum albumin, 5% donkey serum, 0.3 M glycine (Carl Roth GmbH & Co. KG, Germany) and 0.025% Triton X-100 in PBS. We solved primary antibodies in blocking buffer and incubated the cells with them overnight in a 4 °C fridge: (anti-SOX2 (mouse, R&D Systems, USA) 1: 500; anti-NESTIN (rabbit, Merck, USA) 1: 500; anti-TUBB3 (chicken, Merck, USA) 1: 500; anti-GALC (mouse, Merck, USA) 1: 500; anti-GFAP (rabbit, abcam, USA) 1:600; anti-MAP2 (mouse, BD Bioscience, USA) 1: 500; anti-TH (rabbit, Pel Freez biologicals, Rogers, Arkansas, USA) 1: 500; anti SMI-32 (chicken, Covance, Princeton, New Jersey, USA) 1: 10,000). The next day, we washed the cells three times with PBS before secondary antibodies were incubated for 1 h at room temperature in the dark (all Alexa Fluor, 1: 500; Thermo Fisher Scientific, Waltham, Massachusetts, USA). Following this, we washed the cells three times with PBS and a counter-stained with Hoechst 33342 (0.75 µL/mL PBS; Thermo Fisher Scientific, Waltham, Massachusetts, USA) for 5 min. Cells were again washed two times with PBS before we transferred the coverslips onto object slides with Fluoromount (SouthernBiotech, Birmingham, Alabama, USA) as a mounting media. After one day, we sealed the rims of the coverslips with nail polish to prevent excessive drying.

### 4.7. Quantitative Reverse Transcription Polymerase Chain Reaction

We performed qPCR in order to investigate the expression of *NTRK1*,*2* and *3*. We cultivated mature midbrain/hindbrain neuronal cultures of ChAc patients and controls for three weeks in 6-well plates. We washed the cells with pre-warmed PBS and isolated their mRNA using the RNeasy Mini Kit (Qaigen, Hilden, The Netherlands) according to the manufacturer’s instructions. We measured mRNA concentration with a photometer and surveilled the quality by analyzing the 260 nm/230 nm and 260 nm/280 nm absorbance ratios. We then proceeded to generate cDNA from 1000 ng of mRNA using the QuantiTect Reverse Transcription kit (Qiagen, Hilden, The Netherlands) according to the manufacturer’s instructions. qPCR was conducted by using 1 µl of the previously yielded cDNA and the QuantiTect SYBR Green PCR kit (Qiagen, Hilden, The Netherlands). We designed primer-pairs specific for *NTRK1*, *NTRK2* and *NTRK3*. We used 18S rRNA as a reference gene. For more details regarding the primers used, please refer to Table 2. We calculated expression values with the ΔΔ*c*t method. We analyzed each sample at least with technical duplicates.

### 4.8. Western Blot

In order to visualize and quantify the G/F tubulin as well as acetylated tubulin, we used Western blots. We obtained fractioned protein samples as previously described, to yield globular tubulin containing extracts and filamentous tubulin containing ones [34]. In short, we firstly washed two 6-wells of matured neurons with ice-cold phosphatase inhibitor. We then discarded the solution and incubated the cells for 5 min with 200 µL of cold G-actin extraction buffer (0.3% Triton X-100, 5 mM Tris, pH 7.4, 2 mM EGTA, 300 mM sucrose, 2 µM phalloidin, and freshly added Protease Inhibitor Cocktail (1:100)). We collected the soluble fraction stored it at −80 °C, while 200 µL of cold RIPA buffer (1% TritonX-100; 50 mM Tris-HCl; 0.15 M NaCl; 1% Natriumdeoxycholate; 0.1% SDS; 1 mM EDTA; 1 mM DTT; pH 7.4 and freshly added Protease Inhibitor Cocktail (1: 100)) was put in the well. We scraped the cells off the well and collected them in a 1.5 mL reaction tube and incubated them on ice for 15 min. We centrifuged the tube for 2 min at > 10,000 *g* at 4 °C to separate the insoluble fraction and transferred the supernatant into a new tube, while the pellet was discarded. We stored this so obtained F-actin containing fraction at −80 °C. We gained western blots samples for acetylated tubulin from whole cell lysates from maturated neuronal cultures. In short, we washed cells with PBS containing phosphatase inhibitor (Thermo Fisher Scientific, Waltham, Massachusetts, USA), scraped them into 1.5 mL reaction tubes (Eppendorf, Hamburg, Germany) and centrifuged them for 5 min at 200 *g* at 4°C. We then removed the supernatant and resuspended the cell pellet with RIPA buffer, thoroughly mixed the tube and incubated it for 15 min on ice. We then centrifuged the lysate for 5 min at >10,000 *g* at 4 °C and stored the protein containing supernatant at −80 °C, while discarding the DNA containing pellet. We measured protein concentration using Roti Nanoquant (Carl Roth, Karlsruhe, Germany) according to the manufacturers recommendations with a Tecan Sunrise plate reader (Absorbance: 450 nm and 590 nm).

For gel electrophoresis, we loaded same sample volumes per lane—in case of G-tubulin extract and F-tubulin extract—or 15 µg of protein for acetylated tubulin into 12% Bis-Tris precast gels (Life technologies, Carlsbad, California, USA). We conducted electrophoresis at 120 V for one and a half hours. We then blotted the samples onto PVDF membranes (Life Technologies, Carlsbad, California, USA) using an iBlot (Thermo Fisher Scientific, Waltham, Massachusetts, USA; Program 2, 9 min) and a discontinuous buffer condition. We used 25 mM Tris-HCl, 192 mM glycine and 15% methanol as anode buffer and 25 mM Tris-HCl, 192 mM glycine and 0.1% SDS as cathode buffer. We blocked membranes for 2 h with blocking solution (5% Amersham ECL Prime Blocking Reagent, GE Healthcare, Chicago, Illinois, USA). We used mouse anti-alpha tubulin (1: 5000, #7291, Abcam, Cambridge, UK), mouse anti-acetylated tubulin (1: 5000; T7451, Sigma Aldrich, St. Louis, Missouri, USA), hrp-conjugated anti-mouse secondary antibody (1: 5000, Dianova, Hamburg, Germany) and electrochemiluminescence ECL solution (GE healthcare, Chicago, Illinois, USA) for visualization on an LAS 3000 imager (0.8 aperture, 10 s increment, sensitivity high 0.8 aperture, Fuji, Minato, Tokyo, Japan). We performed evaluation of the western blots in the TotalLab Quant software (Totallab, Newcastle-Upon-Tyne, UK). For the quantification of acetylated tubulin, we first developed a membrane with the anti-acetylated tubulin antibody and then proceeded to strip the antibody off by applying two times stripping buffer (1% Tween 20, 0.1% SDS and 1.5% glycine in de-ionized water at pH 2.2) for 1 h. We then washed the membranes twice with PBS for 10 min and twice with PBS-Tween for 5 min. Then we continued by blocking and anti-alpha tubulin incubation procedure as described above. Thereafter, we calculated the ratio of both exposures.

### 4.9. Live Cell Imaging

After the maturation split, we grew the neuronal cultures for a total of three weeks. We used MitoTracker DeepRed, LysoTracker Red DND-99 or JC-1 (50nM, 50nM or 200nM, respectively; all Thermo Fisher Scientific, Waltham, Massachusetts, USA) according to the manufacturer’s recommendation and imaged the cells with a TIRF 6000. We kept exposure time at 115 ms obtaining video stacks at 3.3 fps for acquisition of two channels, 100× 1.4 Zeiss Plan-Apochromat, 400 pictures (2 min duration) hardware synchronized. We used the same illumination intensity and exposure time when taking images of JC-1 incubated cultures in the red and green fluorescence channels. We used self-made Fiji macros to determine the shape of the organelles and “TrackMate” plugin for tracking (settings: Dog detector, blob diameter 1.6 µm, quality threshold 100, kalman tracker, initial search radius 2 µm, kalman radius 1.25 µm, Max Gap distance 2). We used defined fields-of-view of micro channels of MFCs close to the channel entry at the “soma”-site and close to the channel exit at the “synapse”-site. Data extraction and mining was conducted using KNIME.

### 4.10. Statistics

We used GraphPad Prism 7 (Graphpad Software Inc., San Diego, California, USA) for statistical analysis. To evaluate the differentiation capacity and phenotype description of the live cell imaging experiments we calculated Student’s *t* test. When significant differences in the variances between groups were observed, we applied Welch’s correction. For MFC-based live cell imaging experiments as well as PP2 treatment experiments a 2 × 2-way ANOVA was designed with either Šidák’s test to calculate main effects or Tukey’s range test to compare all means.

## Figures and Tables

**Figure 1 ijms-21-01797-f001:**
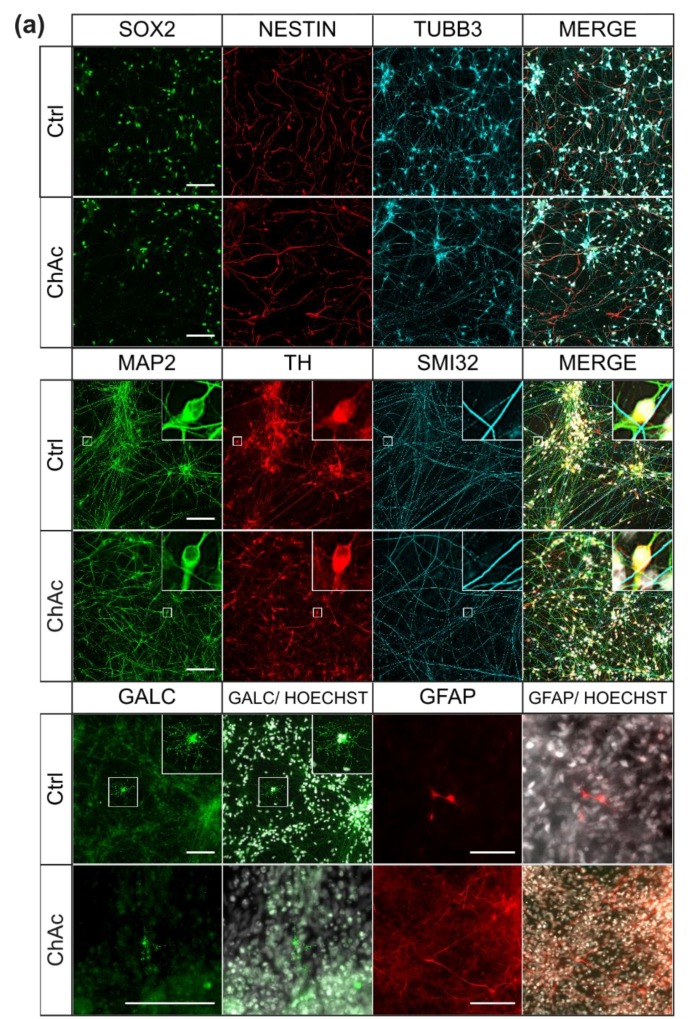
Characterization of the midbrain/hindbrain-differentiation protocol (**a**) Representative ICC images of ChAc patient iPSC derived mature midbrain/ hindbrain stem cell and neuronal cultures after three weeks. All cell lines show cells positive for stem cell/ early neuronal proteins (nestin, SOX2, TUBB3). Stainings for MAP2 and TH indicate the presence of mature dopaminergic neurons. Less than 1% of the cells express GALC or GFAP. (**b**) Quantification of IF images was done in Fiji, using macros which detected nuclei positive for each marker (*n* ≥ 5). (**c**) Size comparison of MAP2^+^ and SMI32^+^ networks. Analysis was done in Fiji by skeletonizing thresholded IF images of MAP2/ SMI32 co-stainings and measuring the length of the obtained networks (*n* ≥ 3). (**d**) Estimation of degeneration of the neuronal network. Fragmentation of MAP2^+^ and SMI32^+^ networks was calculated by assessing the network length and its perimeter and used as a surrogate for neurodegeneration (*n* ≥ 3). (**e**) Decreased survival of ChAc patient iPSC derived mature midbrain/ hindbrain neurons after prolonged (6 months) cell culture. Cell amount was assessed by incubation of neuronal cultures grown in 96-wells with Hoechst 33342 and measuring the fluorescence using a Tecan Genios plate reader (*n* ≥ 6). Scale bar = 100 µm, Boxes represent 25–75 percentiles, line represents median, whiskers represent 10–90%, + represents mean. Bars represent mean ± SEM. **/*** represents *p* < 0.01/0.001.

**Figure 2 ijms-21-01797-f002:**
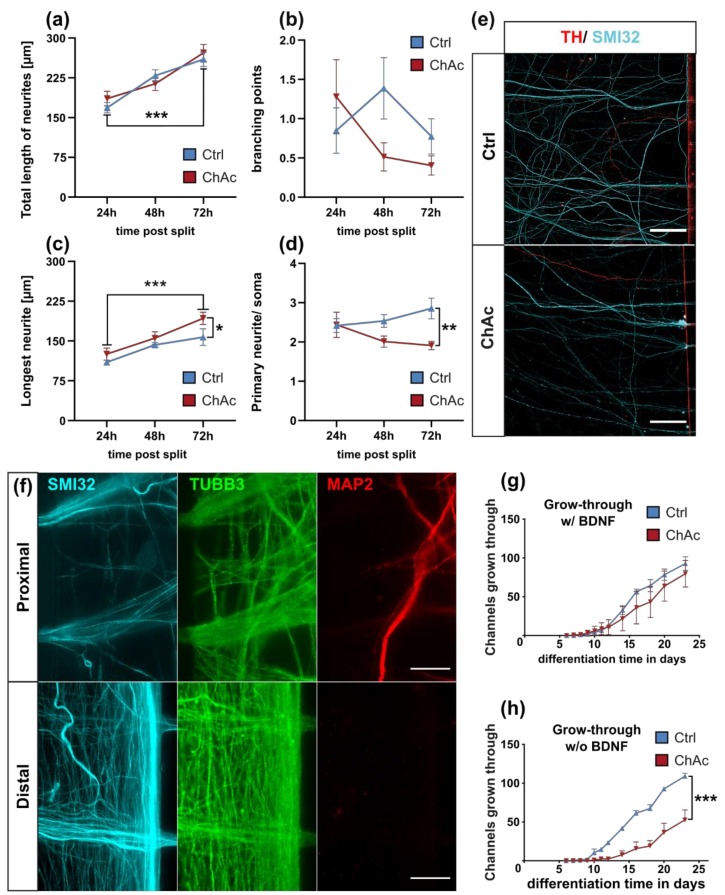
Growth characteristics of differentiated midbrain/ hindbrain neurons. (**a**) The total length of all neurites, (**b**) the number of branching points, (**c**) the longest neurite and (**d**) the number of primary neurites per soma were investigated by IF imaging of individual neurons in low-density cell cultures over the course of 72 h (*n* ≥ 5). (**e**) Representative ICC images of neurons growing through the 900 µm long micro channel of a micro fluidic chamber (MFC). The applied brain-derived neurothrophic factor (BDNF) and glial cell line-derived neurotrophic factor (GDNF) gradient yield mostly TH^-^ neurons, indicating that almost only motoneurons are capable of growing through the channels. (**f**) IF images of axons sprouting out of channel exits of MFCs. Only SMI32^+^ axons penetrate the MFC, while MAP2^+^ dendrites grow not through the micro channels. (**g**) Quantification of channel growth-through in the presence of BDNF was investigated by counting the axon-penetrated exit channels of MFCs (*n* = 3). (**h**) Quantification of channel growth-through in the absence of BDNF (*n* = 3). (**i**) Expression levels of *NTRK1-3* was investigated by qPCR of three weeks old unguided neuronal cultures. Before normalization against *NTRK1*, expression values of each experiment were internally normalized to levels of 18S rRNA (*n* = 15). (**j**) Comparison of the expression levels of *NTRK1-3* of ChAc patient iPSC derived midbrain/ hindbrain neuronal cultures and Ctrl ones. After expression values of each experiment were internally normalized to levels of 18S rRNA, further normalization against Ctrl-values was conducted (*n* ≥ 6). (**k**) Polymerization of tubulin and acetylation of α-tubulin were analyzed using western blots. G/F tubulin ratio was calculated by quantifying western blots of equal volumes of previously extracted globular and filaments containing fractions of three week old neuronal cultures. Acetylation of α-tubulin was measured by first detecting levels of acetylated tubulin and after antibody stripping reanalyzing the same western blot membrane with α-tubulin antibody (*n* ≥ 9). Scale bar = 20 µm, Data points and bars represent mean ± SEM. */**/*** represents *p* < 0.05/0.01/0.001.

**Figure 3 ijms-21-01797-f003:**
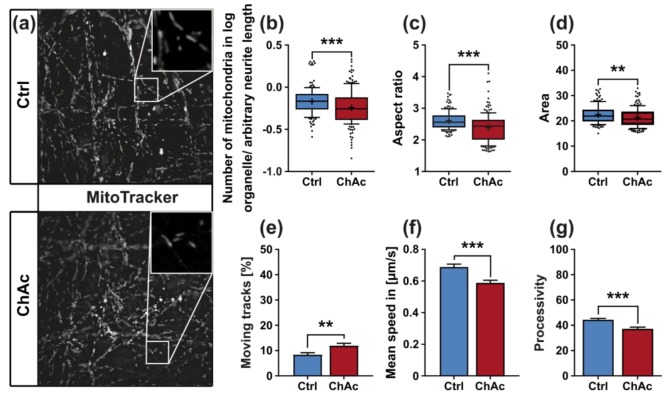
Morphology and trafficking of mitochondria in midbrain/hindbrain neurons cultivated in unguided 96-wells. (**a**) Representative live cell images of mitochondria of ChAc-patient- and Ctrl-iPSC-derived mature midbrain/ hindbrain neurons grown in 96-wells using MitoTracker DeepRed. Analysis of the first image of each video stack with a Fiji macro allowed us to quantify. Magnification 1000x (**b**) the normalized number of mitochondria in neurons (**c**) the aspect ratio of the ellipses that were fitted over the observed mitochondria and (**d**) the pixel-size of mitochondria (*n* ≥ 154). (**e**) Movement of mitochondria was assessed using the TrackMate plugin developed in Fiji and subsequently filtering for objects with a displacement further than 1.2 µm. (**f**) Mean speed was calculated only from mitochondria, which were considered to be moving with a displacement above 1.2 µm. (**g**) Processivity of a moving mitochondrion was calculated by division of its displacement by its total track length. Boxes represent 25–75 percentiles, line represents median, whiskers represent 10–90%, + represents mean. Bars represent mean ± SEM. * indicate significant differences between genotype, **/*** represents *p* < 0.01/0.001.

**Figure 4 ijms-21-01797-f004:**
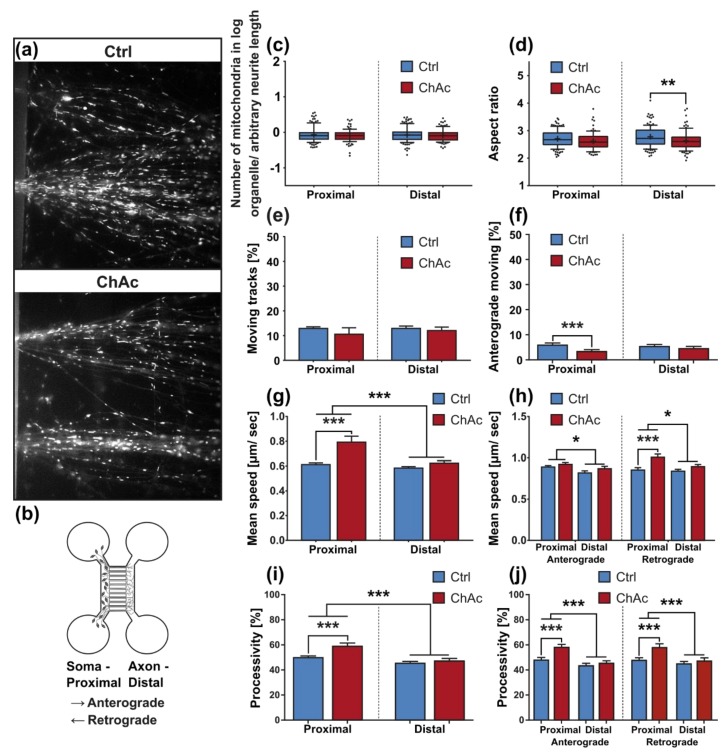
Morphology and trafficking of mitochondria in midbrain/hindbrain neurons cultivated in MFCs. (**a**) Representative live cell images of mitochondria of ChAc-patients and Ctrl-iPSC-derived mature midbrain/ hindbrain neurons sprouting out of the channel exits of MFCs. Mitochondria were visualized using MitoTracker DeepRed. Magnification 1000x (**b**) Schematic of a MFC. Growth-through is guided by applying a positive gradient of BDNF and GDNF. Analysis of the first image of each video stack was conducted with a Fiji macro in defined field-of-views in the micro channels near the channel entries and channel exits of MFCs. This workflow was used to quantify (**c**) the normalized number of mitochondria in neurons and (**d**) the aspect ratio of the ellipses that were fitted over the observed mitochondria (*n* ≥ 94). (**e**) Movement of mitochondria was analyzed over the full video stack in defined fields-of-view in the micro channels near the channel entries and channel exits of MFCs using the TrackMate plugin in Fiji. Tracks with a displacement above 1.2 µm were considered motile (*n* ≥ 19). (**f**) Direction of track vectors was calculated from the track angle parameter obtained from TrackMate plugin from Fiji. Tracks moving from the “soma”-site of MFCs to its “axon”-site were considered as anterograde transport (*n* ≥ 19). (**g**) The mean speed of mitochondria was calculated from TrackMate parameters. Only tracks with a displacement above 1.2 µm were considered in this analysis (*n* ≥ 19). (**h**) A combination of track angle analysis, which was used to determine anterograde or retrograde direction of a track, and mean speed analysis of moving mitochondria was used, to further refine the analysis of mitochondria trafficking (*n* ≥ 19). (**i**) Processivity of mitochondrial movement was analyzed in defined field-of-views in the micro channels near the channel entries and channel exits of MFCs (*n* ≥ 19). (**j**) Analysis of anterograde and retrograde moving mitochondria’s processivity was achieved by dividing tracks based on their track angle (*n* ≥ 19). (**k**) Representative live cell image of the investigation of mitochondria polarization with incubation of JC-1 and subsequent imaging of red (high potential) and green (low potential) fluorescence in ChAc-patients´ and Ctrl neurons in MFCs. Magnification 1000x (**l**) The ratio of red and green fluorescence of JC-1 incubated neuronal cell cultures grown in MFCs was calculated in defined field-of-views in the micro channels near their entries and exits (*n* ≥ 604, individual mitochondria from at least 10 biological experiments per genotype). Boxes represent 25–75 percentiles, line represents median, whiskers represent 10–90%, + represents mean. Bars represent mean ± SEM. * indicate significant differences, */**/*** represent *p* < 0.05/0.01/0.001.

**Figure 5 ijms-21-01797-f005:**
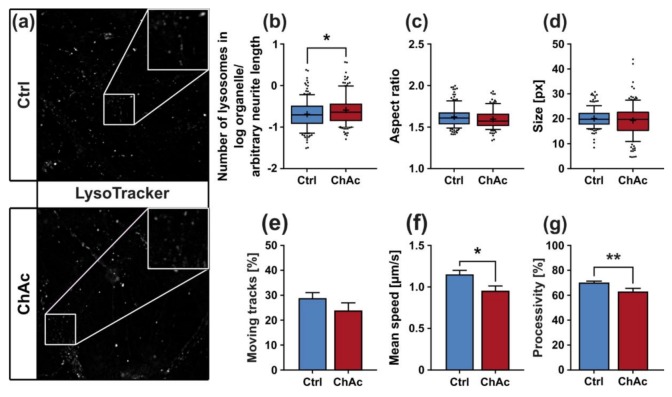
Morphology and trafficking of lysosomes in midbrain/hindbrain neurons cultivated in unguided 96-wells. (**a**) Representative live cell images of lysosomes of ChAc-patients’ and Ctrl-iPSC-derived mature midbrain/hindbrain neurons grown in 96-wells using LysoTracker Red DND-99. Magnification 1000x. The first image of each video stack was analyzed with a Fiji macro to quantify (**b**) the normalized number of lysosomes in neurons, (**c**) their aspect ratio as calculated from ellipses fitted on them and (**d**) their size (*n* ≥ 108). (**e**) Movement of lysosomes was characterized using TrackMate plugin from Fiji. Tracks with a displacement above 1.2 µm were considered motile. (**f**) Tracks, which were considered moving, were further analyzed to calculate their mean speed. (**g**) Processivity of each moving track was calculated by division of its displacement by its total track length. Boxes represent 25–75 percentiles, line represents median, whiskers represent 10–90%, + represents mean. Bars represent mean ± SEM. * indicate significant differences, */** represents *p* < 0.05/0.01.

**Figure 6 ijms-21-01797-f006:**
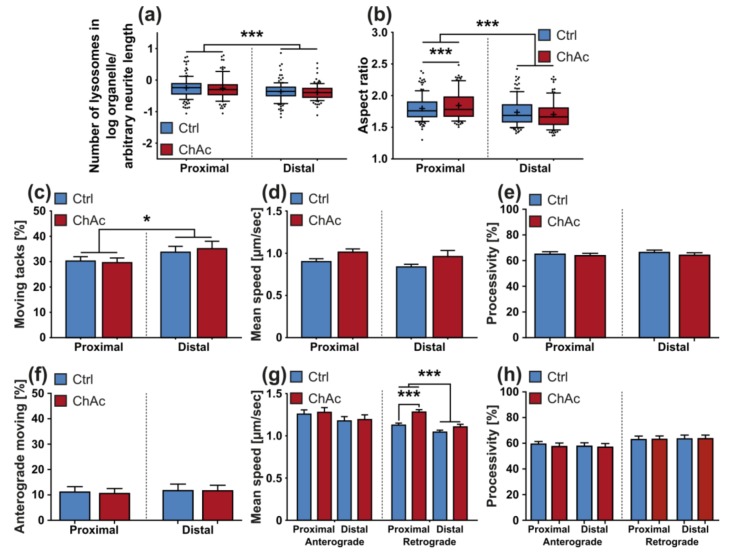
Morphology and trafficking of lysosomes in midbrain/hindbrain neurons cultivated in MFCs. Morphological analysis at defined fields-of-view in the micro channels of MFCs near their entries and exits yielded (**a**) the normalized number of lysosomes and (**b**) their shape (*n* ≥ 90). By differentiating between the defined readout fields-of-view, the trafficking parameters of (**c**) motility, (**d**) mean speed and (**e**) processivity were further dissected (*n* ≥ 19). (**f**) By investigating the track angle provided by TrackMate, the direction of organelle movement was quantified within the defined fields-of-view. (**g**) Separate analysis of the speed of anterograde and retrograde transport of ChAc-patients’ lysosomal vesicles (*n* ≥ 19). (**h**) Processivity of moving lysosomes was analyzed for anterograde and retrograde moving lysosomes individually after assigning them their respective direction using their track angles (*n* ≥ 19). Boxes represent 25–75 percentiles, line represents median, whiskers represent 10–90%, + represents mean. Bars represent mean ± SEM. * indicate significant differences, */*** represent *p* < 0.05/0.001.

**Figure 7 ijms-21-01797-f007:**
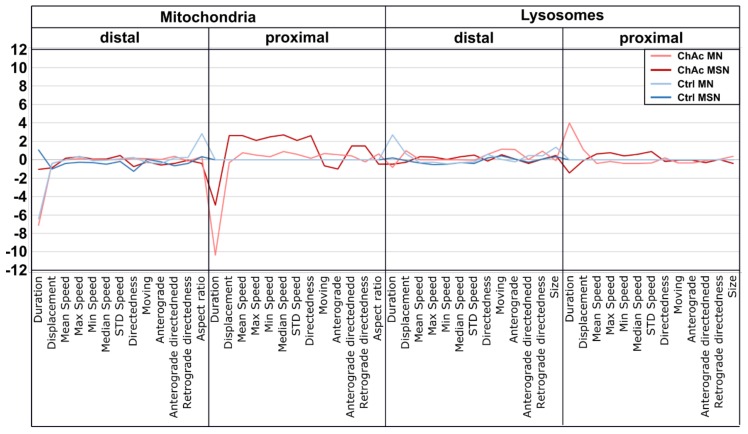
High content disease profile of ChAc-patient’s neurons. Each profile consists of calculated Z-scores compared to their respective control proximal values. Trafficking and morphological parameters of mitochondria and lysosomes in distal and proximal sites were used for profile. Larger deviations from the baseline indicate more severe phenotypes. Data given for ChAc and Ctrl MSN were previously published [21].

**Figure 8 ijms-21-01797-f008:**
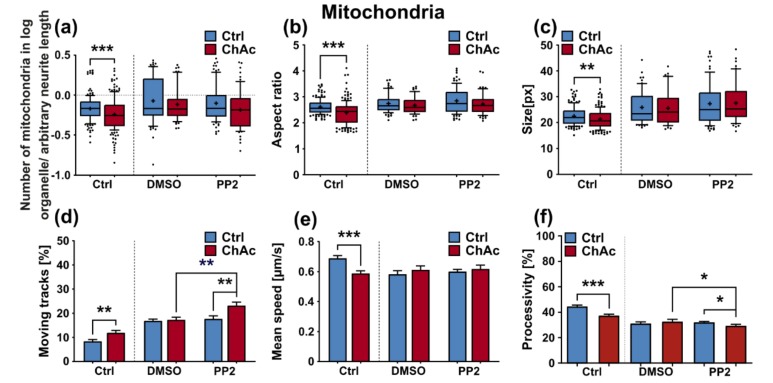
Characterization of morphological and trafficking parameters of mitochondria and lysosomes grown in 96-wells after treatment with the Srk-kinase inhibitor PP2. Morphological analysis with a Fiji macro of live cell images of MitoTracker stained mitochondria yielded (**a**) the normalized number of mitochondria (**b**) the aspect ratio of ellipses fitted on the mitochondria and (**c**) and the size of mitochondria (*n* ≥ 50). Characterization of mitochondria trafficking was achieved by utilization of the Fiji plugin TrackMate. Obtained tracking parameters were (**d**) the moving fraction, which was defined by tracks with a displacement above 1.2 µm, (**e**) and their respective mean speed (*n* ≥ 12). (**f**) Processivity of moving tracks was calculated by dividing their respective displacement by their total track length (*n* ≥ 12). The lysosomal fraction of midbrain/hindbrain neurons cultivated in 96-wells and treated with PP2 was investigated for its morphological parameters as well. (**g**) The normalized number of lysosomes as well as (**h**) their shape and (**i**) size, was calculated from the first image of a video stack from live cell images with LysoTracker by utilizing a macro in Fiji (*n* ≥ 45). The same video stack was then analyzed with TrackMate to yield (**j**) the fraction of motile lysosomes, which was defined by the tracks with a displacement above 1.2 µm (*n* ≥ 12). (**k**) The mean speed of moving lysosomes of neurons grown in 96-wells after treated with PP2 was extracted from the tracking parameters provided by TrackMate (*n* ≥ 12). (**l**) Processivity of the movement of lysosomal vesicles was calculated by dividing each individual tracks displacement by its track length, which were both obtained from the previously conducted analysis with TrackMate (*n* ≥ 12). Boxes represent 25–75 percentiles, line represents median, whiskers represent 10–90%, + represents mean. Bars represent mean ± SEM. * indicate significant differences, */**/*** represent *p* < 0.05/0.01/0.001

**Table 1 ijms-21-01797-t001:** Patient characteristics at the time of skin biopsies. nd = not determined, All lines published by Stanslowsky et al. [20].

Line	Sex	Age at Biopsy	Clinical Features	Molecular Defect	Western Blot for Chorein
ChAc 1	F	31	Chorea, epilepsy, tongue protrusions, lip biting, frontal brain syndrome, peripheral neuropathy	c.4282G > C; c.7806G > A	chorein absent
ChAc 2	F	46	Chorea, epilepsy, peripheral neuropathy	nd	chorein absent
Ctrl 1	F	47	-	nd	chorein present
Ctrl 2	F	53	-	nd	chorein present

**Table 2 ijms-21-01797-t002:** Details of the primer pairs designed for quantification of *NTRK1-3 using qPCR*.

Name	Forward Primer Seq. 5′–3′	Reverse Primer Seq. 5′–3′	Fragment Length (bp)
18S reference	CGT AGT TCC GAC CAT AAA CGA TGC C	GTG GTG CCC TTC CGT CAA TTC C	152
NTRK1	TCT CTC CTT CAA CGC TCT GG	CAC AAG AAC AGT GCA GAG GG	101
NTRK2	TCT GAA CTG ATC CTG GTG GG	CTT GCT GCT TTC ATT CAG GC	126
NTRK3	CGC CAG TAT CAA CAT CAC GG	TGT AGA GCT CCA TGT CCA CG	104

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
