# Peer review of "Combined Dendritic and Axonal Deterioration Are Responsible for Motoneuronopathy in Patient-Derived Neuronal Cell Models of Chorea-Acanthocytosis"

_ijms, 2020, doi:10.3390/ijms21051797_

Round 1
Reviewer 1 Report
Chorea-Acanthocytosis (ChAc) is a rare disease produced by mutations in the gene VPS13A. This work provides very valuable information on the pathophysiology of the disease and the cell types affected.
This is a follow up of a previous article of the same authors showing defects in mitochondria and lysosomes from medium spiny neurons (MSN) differentiated in vitro from patient-derived iPSC. Now the same authors use a different differentiation protocol to study other relevant cell types of the disease: medium dopaminergic neurons and spinal motoneurons. They describe defects if the morphology and trafficking of mitochondria and lysosomes.
I only have a comment regarding the statistical analysis. The authors must carefully revise weather some of the data adjust or not to normal distributions. For example, clear asymmetry can be observed in some data represented by percentiles (see most of the figures), which may indicate lack of normality. If this is the case a non-parametric test should be used to determine the significance (for example the Mann-Whitney test).
Minor remarks
Figure 1. In the middle pannel: I think the second marker that allows a merge is not included or mention.
Description of the boxed-percentile graph must be included in all figure legends.
Lane 311-312. The conclusion described in the text is not concordant with the figure and figure legend.
Author Response
General:
Chorea-Acanthocytosis (ChAc) is a rare disease produced by mutations in the gene VPS13A. This work provides very valuable information on the pathophysiology of the disease and the cell types affected.
This is a follow up of a previous article of the same authors showing defects in mitochondria and lysosomes from medium spiny neurons (MSN) differentiated in vitro from patient-derived iPSC. Now the same authors use a different differentiation protocol to study other relevant cell types of the disease: medium dopaminergic neurons and spinal motoneurons. They describe defects if the morphology and trafficking of mitochondria and lysosomes.
Response: We want to express the reviewer our gratitude for his encouraging and positive view on our work.
I only have a comment regarding the statistical analysis. The authors must carefully revise weather some of the data adjust or not to normal distributions. For example, clear asymmetry can be observed in some data represented by percentiles (see most of the figures), which may indicate lack of normality. If this is the case a non-parametric test should be used to determine the significance (for example the Mann-Whitney test).
Response: We appreciate the reviewers concern regarding our statistical analysis. Indeed, we checked all groups separately and if only one of four groups was not normal distributed—but the underlying biological context suggested normal distribution—, we opted out of applying a non-parametric test in favor of the better understandability of the less convoluted statistical analysis for the readership.
Figure 1. In the middle pannel: I think the second marker that allows a merge is not included or mention.
Response: We agree with the reviewer that this might have been confusing, since we included double and triple ICC stainings in this image. We rearranged the figure 1 appropriately.
Description of the boxed-percentile graph must be included in all figure legends.
Response: During our rework of the figure legends, which were suggested by another reviewer, we added the description of box-blots wherever applicable.
Lane 311-312. The conclusion described in the text is not concordant with the figure and figure legend.
Response: We agree with the reviewer and want to thank him for his thorough study of our manuscript. We corrected the mistake in the figure (missing asterisk) and legend to reflect the appropriate result accordingly.
Reviewer 2 Report
Authors differentiated control and ChAc patients-derived iPSC cells into mature mid-hindbrain neurons. They analyzed fragmentation of neurites using cytochemistry. There was small increase of fragmented network in ChAc patients-derived cell lines when MAP2 marker was used. In prolonged cultures significant decrease of surviving ChAc neurons was observed. Moreover, ChAc neurons show less primary neurites in neurite outgrowth test. Outgrowth of neurites in micro fluidic chambers allowed to selectively observe axons of motoneurons. ChAc motoneurons showed diminished outgrowth in the absence of brain derived neurotropic factor (BDNF). In mid/hindbrain neurons cultures grown in wells authors observed that ChAc- derived cells show reduced number of mitochondria of changed morphology which show more chaotic movement. ChAc motoneurons grown in micro fluidic chambers show similar number of mitochondria as control cells but show different shape in distal sites and increased retrograde speed and directness in proximal sites of neurites. These cells also show hyperpolarization of mitochondria located in distal sites of nerites. Some parameters of lysosomes were also changed in ChAc mature neurons grown in wells and in motoneurons grown in micro fluidic chambers. Finally, authors studied the effect of Lyn kinase inhibitor on mitochondrial and lysosomal parameters of control and ChAc mature neurons grown in wells and they concluded that there was no effect. However, the effect of DMSO alone (solvent used to dissolve inhibitor of Lyn kinase) was significant making, in my opinion, the results not conclusive.
Authors present many new and significant results. However, the manuscript is difficult to read, contains many errors, mistakes, contains too many statistical parameters in the text of Results. Results shown in Figure 8 are not conclusive, should not be used in the title of the manuscript. Figures are not consistently formatted. English should be improved. Discussion could be improved. List of sggestions is below.
Abstracts
20, VPS13A gene should be in italics.
Chorein, no capital letter in the middle of the sentence in Abstract and in all other text. The same as other proteins: tubulin, actin etc.
21, parkinsonism
24, Lyn-kinase is a target, not Lyn-kinase inhibition
28, number of lysosomes, not amount
Introduction
43, VPS13A, genes in italics
55, Vps13
74, what is PP2? Add to abbreviation list.
Results
Unify p<0.001 (line 162 for example) and p<.0001 (line 184 for example). Why to give so long numbers if they are not significant, for example p=0.2348. SD in the text are not necessary, only in Figure, they disturb reading. Too many statistical parameters shown without explanation what they are. p alone definitely is sufficient. Long line of parameters in brackets on the end of the chapter 2.4 is not helping. On the end of the chapter should be the conclusion. All these long lines of numbers disturb reading and make finding the message more difficult. Is difficult to understand what was compared, for example chapter 2.7.
In many sites information in Results is different than can be shown in the Figure or is described in the legend. It is difficult to know from this what is the message. Looks that authors are lost what was compared to what and what is different.
Describing results it is better to use passive voice, not the mix of active and passive voice sentences.
Add space after the numbers before the unit, 10 µM (% is an exception).
82 in Chapter 2.1 is not clear from which cells differentiation was started. We know this from Abstract.
83, VPS13A
85, genes are expressed, not proteins. Expression of genes was investigated by detecting their protein products?.
86, which markers identify particular cells? What they are?
87, and motoneurons?
95, anti-SMI32 antibody
102 and 106, TH is not in the Figure 1d.
108, p-value?
139, was not affected, no significance on 2b
142, in the text: ChAc neurons had significantly less primary neurites(2d). The significance is shown in 2d. However in the legend is: the number of primary neurites per soma was not different when ChAc … neurons were compared to Ctrl. Remove from the legend what is different, what is not. This should be only in results.
what is the survival after 72 h?
164, out growth or outgrowth, as in 135
174, what is BDNF? Abbreviation should be introduced when first appears.
183, mess of genes and proteins. What was studied? Expression of genes or the level of proteins? Which method? RT-PCR or Western blot?
188, how many repeats? More repeats may help to have better p parameter.
189, p= 0.057 is not significant.
191, Chapter 2.5 without the Figure should not exists as separate chapter. Add (Figure 2k). Alternativewly, negative results can be only mentioned in Discussion.
203, 205 avoid expressions: we wanted, we decided. Researchers make decision all the time.
205 and many other sites, number of mitochondria, number of lysosomes
204 outgrowth
242, what are normal cultures? Cells grown in wells? Normal for neurons is to grow in tissue.
247, can be attributed to motoneurons only in distal part
255, direction of mitochondrial movement.
258 where the sentence ends?
268, “processivity” in the text and “directness” in Figure 4 (i) and (j).What is processivity and what is directness? Why not the same word in the text, figure and legend?
312 is: “significantly increased amount (number) of lysosomes (5b)”, while: “no significant difference shown in Figure 1(b)” and “no differences …(b) amount (number) of lysosomes”.
332 distal sites of what? Axons?
333 morphological analysis of what? Mitochondria?
342 “velocity .. was increased” but no significance in Figure 6D and “no differences” in the legend
335 in distal sites of axon. Do not mix compartments of the chamber with neurite (or axon) sites. These sites are not compartments.
338, what is the aspect ratio?
349 distal site or part of axon.
353 organelles were moving. Past tense when describing results.
356, lysosomes in neurons not in MCFs.
373, 2.10. Wrong title. No results for other diseases are shown below.
374, phenotypes of what?
375, parameters of what?
376, what is MSN? Explain abbsreviation when first appears.
377, distinct profile
394, 2.11 phenotypes of ChAc neurones
399, trafficking of what?
404, moving fraction (Figure 8d). We observed……. in Ctrl ones (Figure 8d).
408, velocity of moving organelles was not different (Figure 8e),
413, it should be mentioned that DMSO alone affects movement of mitochondria
426, characteristics of mitochondria and lysosomes?
Figure 1
Different enlargements for Ctrl and ChAc, why? It is difficult to compare.
Two different signs are used to indicate statistical significance in (d), * and #. It is not clear which parameters were compared. Usually square brackets are used to point what was compared , as is properly shown in Figure 2(i). Legend to Figure 1 also did not help, no description of both signs, what they mean.
(d) it is difficult to believe that the difference for MAP2 is significant, three repetition only.
Legends to all Figures should contain description of the experiment, what is on the figure, materials, controls, statistical analysis (test used) but should not contain description of results (what was higher, what was lower) conclusions and discussion. Sentences “We rarely observed GALC oligodendrocytes” and “supposedly a first sign for neurodegeneration” are like taken from the main text not from the legend. Other legends are also not properly written.
Placing (b), (c), (d), (e) do not obey the rules where the numbers or letters for individual parts should be: outside of the figure or its part. According to the MDPI format should be below the figure part. Upper right corner is usually used but definitely not inside the figure.
In all Figures [%], [µm], [µm/s] brackets should be added. s for seconds.
Why TUJ1+ in (b) while TUBB3 in the text, line 85?
Figure 2
According to Authors instructions and format Figure cited in the chapter should be placed below the chapter. Figure 2 is a combination of three Figures for three chapters. Figure is lacking images of cells which were analysed in (a) to (d).
Wrong format, different than in Figure 1. Letters (a), (b) are now in bold. Again two kind of signs for statistical significance in (d), * and #. Why to use two tests to find significance of one result? One is sufficient. Makes all less clear.
Description of images should be unified in all figures but here is different in (e) and (f). Letters in (i), (j), (k) should not be bold and size of letters should be the same in all parts and all figures of similar type. In (h) results compare should be pointed by brackets on the right.
In (i) the expression of genes NTRK1,2,3 was shown (text line 183) but, curiously, the protein TrkA was taken as control but not house-keeping gene, like gene encoding actin. It is not written what was the method of analysis of gene expression. Moreover, in the Figure 2 legend (line158) NTKR1-3 are described as receptors. It is really difficult to know how the experiment was performed. Mix of genes and proteins. Maybe authors studied the level of TrkA, TrkB and TrkC proteins? They still need external control.
Space should be added between the number and unit, 900 µm.
How many times experiment shown in (j) was repeated?
Figure 3
Wrong title. Mitochondria were in neurons, not in wells. Mitochondria were in neurons which were grown in wells.
Images (a) contain large field, we do not see morphology of mitochondria. Window with enlarged fragment of the field is needed.
Figure 4
Wrong title. Mitochondria were not trafficking in chambers. Mitochondria were in neurons.
Why * and # ? Brackets are clear to show what was compared. Figures should be self-explanatory, as much as possible.
Figure parts should not overlap, like (h) and (j).
In the legend and in the mail text authors mix what is the compartment. Compartment is the part of the chamber, not the proximal or distal part (or site) of the axon or neurite. Subcellular compartments are surrounded by membranes, like mitochondria or lysosomes. Parts of neurite are not separated by membrane.
278, Elongated in the distal site, according what is in d. Again, the reader will see alone what is less and what is the same. Instead, is better to explain in the legend what is an aspect ratio.
number of moving mitochondria, not amount. Also wrong in other sites.
287,288, wrong sentence , movements of mitochondria.. on… mitochondria
291 hyperpolarization when in the distal compartment of MFC. Neurons do not have MFC.
Figure 5
Wrong title as in Figure3/4.
322 neurons grown in wells
No active voice in the legend: we observed ..
Figure 6
Wrong title.
362 the number of lysosomes
Wrong legend. What was analysed, not what was higher, what was lower, what was faster. Reader will see alone what is the result.
Figure 7
Font in right upper window is too small.
391 wrong sentence: intensity …is less intense
Description what is less intense and what is striking should be in Results, not in the Figure legend. Here only what is shown in the Figure.
Figure 8
429, Lysosomes and mitochondria, not all organelles. Remove 48 h from the title and transfer to the legend.
Statistical significance in Ctrl should be shown, as is in Figure 3.
DMSO is alleviating the difference between wild type and ChAc cells in many parameters.
Many figure parts are not cited in the text.
Figure A1
(b) section/perimeter =2, but in the legend perimeter/length of 2 of 2. How this can be? The same mistake in (c).
Discussion
445, ChAc abbreviation was already introduced and should be used to the end of the manuscript.
454, which is in agreement
468, phenotype is always present, which phenotype, as wild type or different.
468, the level of TrkA, B and C was not different
471, documented before
476, Here we found
477, with mitochondria in MSN. The trafficking phenotype of mitochondria was similar. What pattern?
What is the conclusion from hyperpolarisation of mitochondria which was observed in ChAc cells?
485 decrease in number of organelles (lysosomes) was not shown in these studies. In Figure 5b we don’t see this.
498, Conclusion not supported be data in Figures, add reference for phenotype of ALS. Add references in the paragraph.
504, Hyperpolarisation of mitochondria was observed in distal part of neurite (Figure 5). If this is the sign of damage of mitochondria it means that distal part of neurite degenerates first, not proximal as is stated.
506, If DMSO shows significant effects should not be used.
Remove concentration from Discussion.
510, by parts of signalling pathway?
Materials and methods
Why clones derived from two different patients were mixed? It is known, they are similar?
Mix of active and passive voice in one paragraph.
Reviewer 3 Report
The authors characterized spinal motor neuron cultures differentiated from Chorea Acanthocytosis patient-derived iPSCs. They found similar phenotypes as the ones they previously described in patients iPSCs-derived medium spiny neurons, namely alteration in the size and trafficking of mitochondria and lysosomes in the axon. Furthermore they report that these phenotypes are not rescued by inhibition of the Lyn kinase, which was previously reported to have ameliorating effects on other Chorea Acanthocytosis disease cell models.
The study uses high quality technical approaches, including differentiation of patients-derived iPSCs into specific neuronal subtypes and microfluidic chamber to allow selective studies of axonal organelles in the distal or proximal compartment. Experiments are carried out with appropriate controls, quantification and statistical analysis. However, the overall impact of the results is rather weak. Results are not linked together by any mechanistic evaluation nor they are tied back to the cellular function of Chorein.
Additionally, a whole portion of relevant literature on the basic function of VPS13 proteins and of Chorein specifically (which was shown to be a lipid transport protein localized at membran contact sites between mitochondria and endoplasmic reticulum) have been almost completely overlooked in both the introduction and in the discussion and interpretation of the results.
Several grammatical errors need corrections and the list of references needs to be carefully checked for gross mistakes.
Author Response
General:
The authors characterized spinal motor neuron cultures differentiated from Chorea Acanthocytosis patient-derived iPSCs. They found similar phenotypes as the ones they previously described in patients iPSCs-derived medium spiny neurons, namely alteration in the size and trafficking of mitochondria and lysosomes in the axon. Furthermore they report that these phenotypes are not rescued by inhibition of the Lyn kinase, which was previously reported to have ameliorating effects on other Chorea Acanthocytosis disease cell models.
The study uses high quality technical approaches, including differentiation of patients-derived iPSCs into specific neuronal subtypes and microfluidic chamber to allow selective studies of axonal organelles in the distal or proximal compartment. Experiments are carried out with appropriate controls, quantification and statistical analysis. However, the overall impact of the results is rather weak. Results are not linked together by any mechanistic evaluation nor they are tied back to the cellular function of Chorein.
Response: We want to thank the reviewer for his overall positive feedback on our work. We agree with the reviewer that our work does not finally depict the mechanistic pathway. However, we highlight that the motoneuron phenotypes are not caused by Lyn kinase pathway, which is somehow surprising taking into account the multiple results of other ChAc disease models. Thus, we believe that our manuscript will help the scientific community to gain a better understanding of the disease itself as well as the even less explored affected tissue, including motoneurons. Of course, further studies are required, that have to investigate the mechanistic links of the herein presented phenotypes.
Additionally, a whole portion of relevant literature on the basic function of VPS13 proteins and of Chorein specifically (which was shown to be a lipid transport protein localized at membran contact sites between mitochondria and endoplasmic reticulum) have been almost completely overlooked in both the introduction and in the discussion and interpretation of the results.
Response: We agree with the reviewer that our introduction to the topic was pretty focused and thus might have missed some advances in the field. We extended the paragraph, which was explaining the functional aspects of VPS13A—including reference to ERMES—by most recent publications. We focused on the function of chorein as lipid transporter and its localization at membrane contact sites especially in the ER and on mitochondria.
Several grammatical errors need corrections and the list of references needs to be carefully checked for gross mistakes.
Response: We revised most chapters regarding grammatical and orthographical mistakes as well as being more consistent with active and passive voice in the Results and Material and Methods sections. Gross mistakes in the list of references have been corrected and the list in general was updated with most recent publications regarding VPS13A and ChAc research. We think that after the revision our manuscript is accurately reflecting the current state of knowledge as much as it can be depicted in an original work manuscript.
Round 2
Reviewer 2 Report
Manuscript was greatly improved and is now suitable for publication.